# Facile Asymmetric Syntheses of Non-Natural Amino Acid (*S*)-Cyclopropylglycine by the Developed NADH-Driven Biocatalytic System

**Qian Tang [1], Shanshan Li [2], Liping Zhou [2], Lili Sun [2], Juan Xin [2] and Wei Li [2,*]**

[1] School of Pharmacy, Chongqing Medical and Pharmaceutical College, 82 Middle College-City Road, Chongqing 401331, China; tangqian@cqmpc.edu.cn
[2] Department of Medicinal Chemistry, School of Pharmacy, Chongqing Medical University, 1 Yixueyuan Road, Chongqing 400016, China
* Correspondence: li_wei@cqmu.edu.cn; Tel./Fax: +86-23-6848-5161

**Abstract:** A self-sufficient bifunctional enzyme integrating reductive amination and coenzyme regeneration activities was developed and successfully employed to synthesize (*S*)-cyclopropylglycine with an improved reaction rate 2.1-fold over the native enzymes and a short bioconversion period of 6 h at a high substrate concentration of 120 g·L$^{-1}$ and space–time yield of (*S*)-cyclopropylglycine up to 377.3 g·L$^{-1}$·d$^{-1}$, higher than that of any previously reported data. Additionally, (*S*)-cyclopropylglycine could be continuously synthesized for 90 h with the enzymes packed in a dialysis tube, providing 634.6 g of (*S*)-cyclopropylglycine with >99.5% ee and over 95% conversion yield up to 12 changes. These results confirmed that the newly developed NADH-driven biocatalytic system could be utilized as a self-sufficient biocatalyst for industrial application in the synthesis of (*S*)-cyclopropylglycine, which provides a chiral center and cyclopropyl fragment for the frequent synthesis of preclinical/clinical drug molecules.

**Keywords:** (*S*)-cyclopropylglycine; bifunctional enzyme; coenzyme regeneration; reductive amination; asymmetric synthesis

## 1. Introduction

Polypeptide, as a natural messenger, plays an important role in various biological processes. Therefore, peptide-based drugs are often utilized to treat diabetes, hypertension, and osteoporosis, with fewer side effects. In particular, the demand for these drugs is likely to continuously increase [1]. However, some issues such as poor selectivity, biodegradation, and high conformational flexibility seriously hamper their clinical application [1]. To resolve these issues, the introduction of non-natural amino acids is a useful alternative. In recent decades, non-natural amino acids have often been incorporated into various bioactive peptides such as cyclosporins, vancomycin and actinomycins, consequently leading to improved bioavailability and metabolic stability [1–3]. As a representative example, (*S*)-cyclopropylglycine is of considerable interest due to it being a biomedically active compound and a vital building block to synthesize the peptide and peptidomimetic family, members of which have therapeutic and commercial value (Figure 1), such as macrocyclic ghrelin receptor agonists [4], kinesin spindle protein inhibitors [5], TRPA1 receptor antagonists [6], and iminohydantoin BACE1 inhibitors [7]. Due to its multifunctional groups, (*S*)-cyclopropylglycine can provide a chiral center and a cyclopropyl fragment for the synthesis of preclinical/clinical drug molecules [8]. Therefore, an efficient method of asymmetric synthesis of (*S*)-cyclopropylglycine remains essential.

**Figure 1.** Some examples of active compounds containing (*S*)-cyclopropylglycine.

The current chemical and biochemical strategies to produce (*S*)-cyclopropylglycine rely on the Strecker reaction [9], or papain-catalyzed deracemization of racemic methyl N-Boc-cyclopropylglycinate [10], or Pd$^{II}$-catalysed cyclopropanation of ethyl-2-tert-butoxycarbony-lamino-2-cycloppylethanoate [11]. However, these methods require toxic reagents (KCN) and hazardous solvents, expensive starting materials, and only provide a low yield. Meanwhile, an enzyme-catalyzed kinetic resolution has been carried out to produce (*S*)-cyclopropylglycine, but it only provided <50% yield [10,12]. Notably, biocatalytic approaches to chiral non-natural amino acid synthesis were explored to improve a tom economy and offer a cleaner and greener alternative. Among them, the reductive amination catalyzed by amino acid dehydrogenases was investigated to produce (*S*)-cyclopropylglycine, due to its good product stereoselectivity, lack of by-product, and high equilibrium constant. Parker et al. described an enzyme-coupled system (Figure 2a) consisting of leucine dehydrogenase and formate dehydrogenase to furnish the reductive amination of cyclopropyl-glyoxylic acid with NADH cofactor recycling for the production of (*S*)-cyclopropylglycine at substrate concentrations of about 100 g/L, providing 99% yield and 100% ee after 17 h of reaction [13]. Although the enzyme-coupled strategy was attractive, owing to it being a one-pot route to reductive amination and cofactor regeneration, a considerable number of issues, such as a prolonged bioconversion period, the need for two separate enzymes, and recycling and reuse, remain. The success in DNA recombination technology has instigated us to seek a new biocatalytic strategy to synthesize (*S*)-cyclopropylglycine.

As an attractive alternative to using two separate enzymes, genetic fusion is the simplest but most efficient strategy to allow two open reading frames that encode different active sites to be brought into close proximity [14–17]. This thereby forms substrate channeling which can speed up intermediate turnover in the consecutive reaction by reducing the transit time required for the intermediate to reach the enzyme, which catalyzes the next-step reaction by ensuring that a high local concentration of intermediate exists in the vicinity of the active sites of this enzyme, and also by significantly limiting the diffusion of intermediates [18,19]. This consequently provides a higher titer and yield, and faster reaction rates over the free subunit. For example, L-glutamate-γ-semialdehyde dehydrogenase (GSALDH) and proline dehydrogenase (PRODH), encoded by separate genes, were covalently combined into one polypeptide chain, producing a bifunctional enzyme proline utilization A (PutA) which catalyzed the two-step oxidation of proline to glutamate, and the high activity of PutA was best described by the substrate channeling mechanism [20–22]. Albertsen et al. constructed the fusion protein (FPPS-GSG-PTS) of patchoulol synthase (PTS) and farnesyl diphosphate synthase (FPPS), and a more than 2-fold improvement in patchoulol titer was obtained compared with the separate enzymes [19]. Yu et al. also demonstrated that the expression of fusion protein 4CL (4-coumarate-CoA ligase)::STS (stilbene synthase) increased resveratrol production up to 15-fold compared to the coexpression of 4CL and STS [23]. Impressively, Iturrate et al. described an engineered bifunctional aldolase/kinase enzyme which conferred a catalytic advantage for C-C bond formation re-

sulting in a 20-fold increase in the reaction rates over the parent enzymes [24]. Additionally, a key point was that the number of enzymes that participated in the biocatalytic reaction as well as were needed to purify was reduced when a fusion enzyme was used. Considering these possible advantageous effects, we also successfully employed the fusion enzyme to produce (*R*)-3-quinuclidinol in a high space–time yield [25]. To the best of our knowledge, there is no report on the fusion enzyme as a biocatalyst employed to synthesize unnatural amino acid (*S*)-cyclopropylglycine to date.

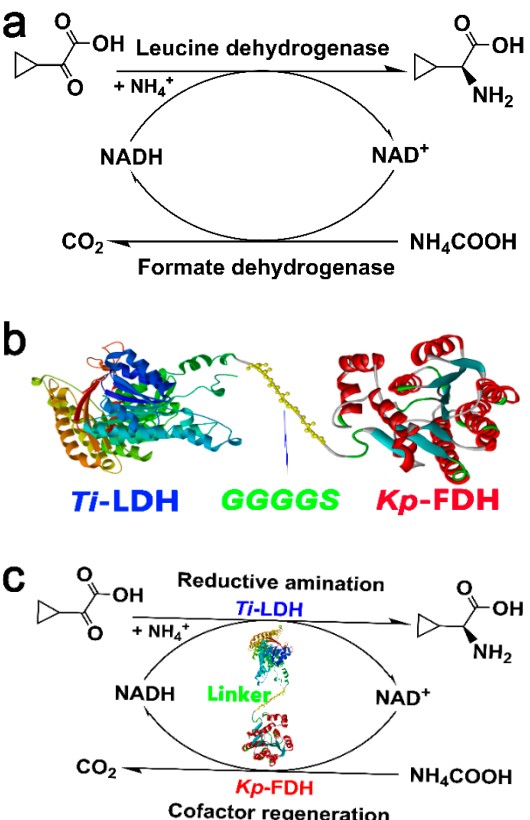

**Figure 2.** (**a**) Enzyme-coupled biocatalytic process for the synthesis of (*S*)-2-cyclopropylglycine and cofactor regeneration. (**b**) Construction of bifunctional enzyme, and (**c**) Bifunctional enzyme-mediated asymmetric synthesis of (*S*)-2-cyclopropylglycine and cofactor regeneration.

The biocatalytic process often requires the stoichiometric addition of the expensive coenzyme NADPH to provide reduction power, which is undesirable in large-scale production. Although coenzyme regeneration in situ has been extensively investigated, a catalytic amount (0.1–0.5 mM) of NADP$^+$ should be added into the biocatalytic system to achieve effective cofactor regeneration [26]. In comparison with expensive NADP$^+$, the price of commercialized NAD$^+$ was only 25% of that of NADP$^+$ (USD 2000/kg) [26]. Thus, the development of the NADH/NAD$^+$-driven biocatalytic system is highly desired. The glucose dehydrogenase-catalyzed coenzyme regeneration system has often been employed to regenerate enzymes. However, the produced by-product gluconic acid leads to difficulty in subsequent product separation. The formate dehydrogenase (FDH)-catalyzed coenzyme regeneration system is more favored owing to its general availability, benign by-product ($CO_2$), and low reduction potential [26].

Encouraged by these facts, here, a bifunctional enzyme was engineered to develop an NADH-driven biocatalytic system to produce (*S*)-cyclopropylglycine, in which reductive amination and coenzyme regeneration can run concurrently. Therefore, bifunctional enzymes were first constructed by fusing *Thermoactinomyces intermedius* (EC:1.4.1.9) leucine dehydrogenase (*Ti*-LDH) and *Komagataella pastoris* (EC:1.17.1.9) formate dehydrogenase

(*Kp*-FDH) by linker peptide (GGGGS) (Figure 2b). The NADH-driven biocatalytic system successfully catalyzed the asymmetric syntheses (*S*)-cyclopropylglycine in two successive reactions, in which cyclopropylglyoxylic acid underwent a reductive amination to (*S*)-cyclopropylglycine, and the NADH cofactor was oxidized in the *Ti*-LDH active site, while ammonium formate underwent oxidation in the *Kp*-FDH active site in which a hydride was transferred to NAD$^+$, thus regenerating NADH (Figure 2c).

## 2. Results and Discussion

### 2.1. Expression and Purification of Recombinant Enzymes

Wild-type leucine dehydrogenase from *Thermoactinomyces intermedius* (*Ti*-LDH) and formate dehydrogenase from *Komagataella pastoris* (*Kp*-FDH) were chosen and then integrated into the bifunctional enzyme by a linker GGGGS, namely TLK, which contained a *Ti*-LDH module in the N-terminal of the polypeptide chain and a *Kp*-FDH module in the C-terminal, and KLT in reverse connection *Ti*-LDH and *Kp*-FDH. Plasmids were transferred into competent *E. coli* BL21(DE3) to form the engineering bacteria *E. coli* (*Ti*-LDH), *E. coli* (*Kp*-FDH), *E. coli* (KLT), and *E. coli* (TLK), respectively. These engineering bacteria were directly used for the biocatalytic process. Preliminary experiments demonstrated that *E. coli* (TLK) exhibited much higher biocatalytic activity toward potassium cyclopropylglyoxylate than *E. coli* (KLT) and a combination of *E. coli* (*Ti*-LDH) and *E. coli* (*Kp*-FDH)) (Figure S1). Therefore, only the bifunctional enzyme TLK was used to perform subsequent experiments. Judging by SDS-PAGE images (Figure 3), it was found that *E. coli* (*Ti*-LDH) and *E. coli* (*Kp*-FDH) expressed a high level of *Ti*-LDH (40.586 KDa) and *Kp*-FDH (40.310 KDa) in a soluble form, while *E. coli* (TLK) produced a slightly low level of TLK bifunctional enzyme due to its large apparent molecular weight which was estimated to be about 81 kDa, in good agreement with the sum of the molecular weight of parent enzymes. To improve target enzyme activity, expression conditions including IPTG concentration, induction temperature and time were optimized. The optimal conditions as well as the corresponding activities are listed in Table 1. TLK exhibited both reductive amination and coenzyme regeneration activity, indicating that the bifunctional enzyme TLK can be used as a self-sufficient whole-cell biocatalyst for the reductive amination of the carbonyl group and cofactor regeneration in situ.

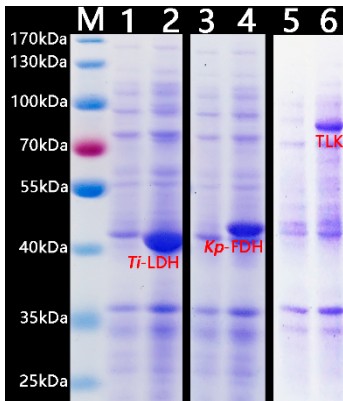

**Figure 3.** 12% SDS-PAGE analysis of *Ti*-LDH, *Kp*-FDH, and TLK expression. Lane M, protein molecular weight markers. Lane 1, 3, 5, and Lane 2, 4, 6, the *E. coli* cell lysate induced before and after 0.4 mM IPTG for *E. coli* (*Ti*-LDH), *E. coli* (*Kp*-FDH), and *E. coli* (TLK), respectively.

**Table 1.** The optimal expression conditions and specific activity of each enzyme.

| Enzyme | Optimal Expression Conditions | | | Specific Activity (U·g$^{-1}$) | |
| --- | --- | --- | --- | --- | --- |
| | Temperature | Time | IPTG | Reductive Amination | Coenzyme Regeneration |
| *Ti*-LDH | 25 °C | 24 h | 0.4 mM | 2086 | / |
| *Kp*-FDH | 25 °C | 48 h | 0.2 mM | / | 398 |
| TLK | 16 °C | 48 h | 0.4 mM | 659 | 145 |

### 2.2. The Effect of Temperature and pH on the Recombinant Enzyme Activity

The effects of temperature and pH on *Ti*-LDH, *Kp*-FDH, and TLK were tested over 20 °C to 60 °C and 5.0 to 10.0, respectively. The optimized temperature for both parent enzyme *Kp*-FDH (Figure 4a) and *Ti*-LDH (Figure 4b) was 35 °C and 40 °C, respectively. Meanwhile, the cofactor regeneration subunit (*Kp*-FDH domains) activity (Figure 4c) and the reductive amination subunit (*Ti*-LDH domains) activity (Figure 4d) in the fusion enzyme TLK was best at 40 °C and 30 °C, respectively. The results were in line with the later-observed thermal stability. It was found that the parent enzyme *Kp*-FDH exhibited better pH tolerance ranging from 7.0 to 9.0 (Figure 4e), while *Ti*-LDH had the highest activity at pH 9.0 (Figure 4f), in line with the reported data [13]. After fusion, the highest enzyme activity for both cofactor regeneration (*Kp*-FDH domains) and reductive amination (*Ti*-LDH domains) in TLK was at pH 8.0 (Figure 4g,h). At pH 9, both of them still kept about 90% relative activities. Thus, the TLK-catalyzed asymmetric synthesis was run at 35–40 °C and pH 8.0–9.0. After 24 h of incubation at different temperatures, the enzyme residual activity was investigated to evaluate the thermostability. As shown in Figure 5, all enzymes exhibited reduced activity with the increase in the temperature and incubation period. *Kp*-FDH domains and *Ti*-LDH domains in the fusion enzyme TLK showed similar thermostabilities to the parent enzymes.

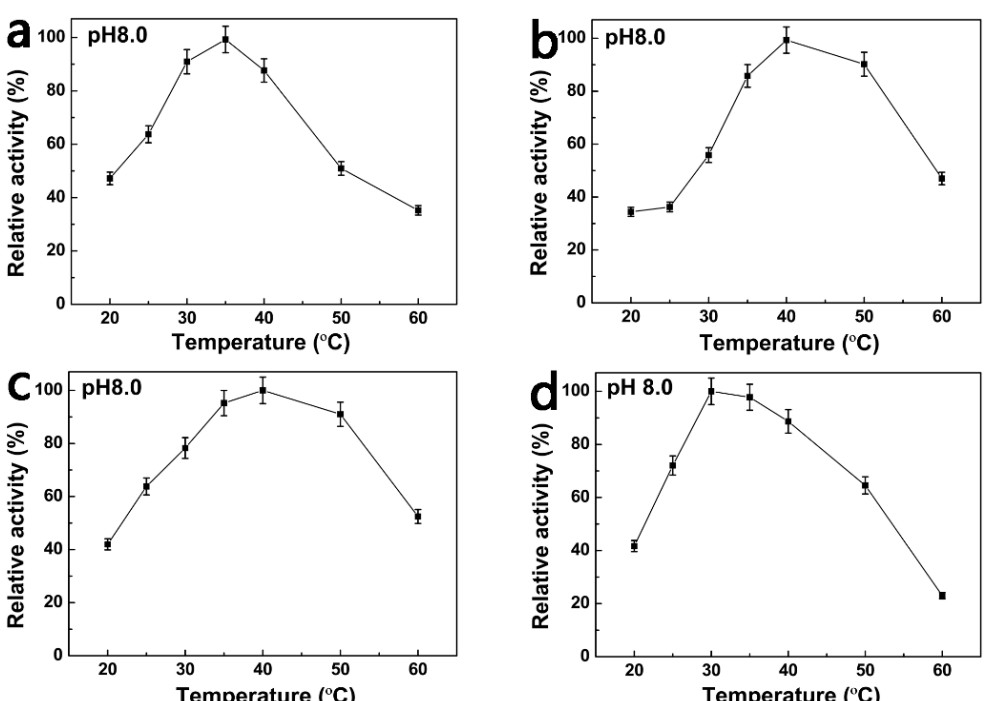

**Figure 4.** *Cont.*

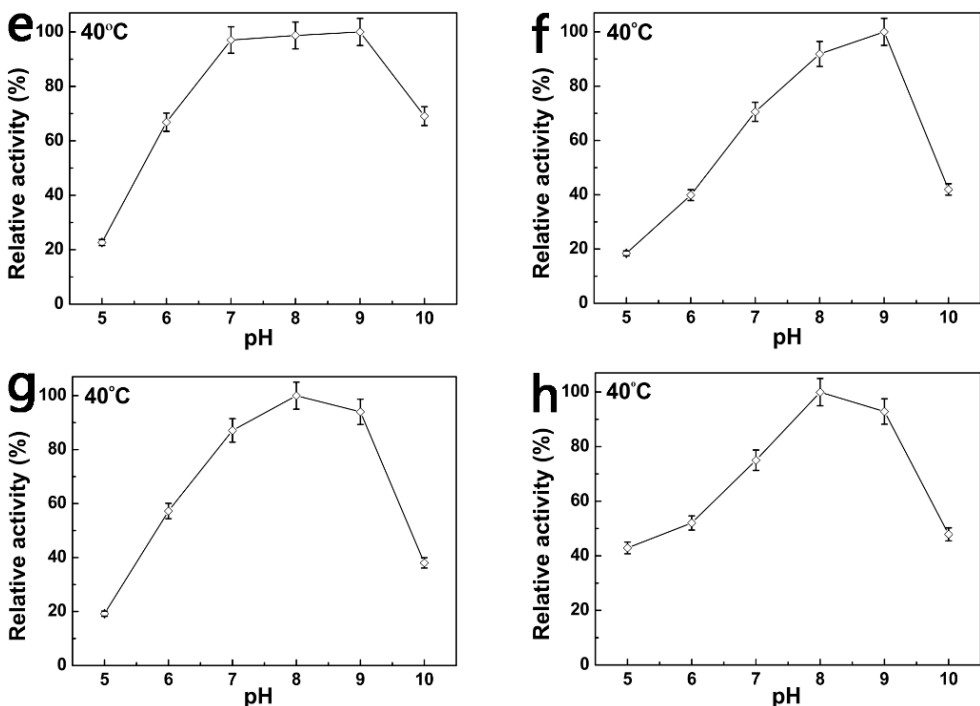

**Figure 4.** The effects of temperature and pH on the activities of *Kp*-FDH (**a**,**e**), *Ti*-LDH (**b**,**f**), cofactor regeneration enzyme (*Kp*-FDH domains) (**c**,**g**) and carbonyl reductive amination (*Ti*-LDH domains) (**d**,**h**) in the bifunctional enzyme TLK.

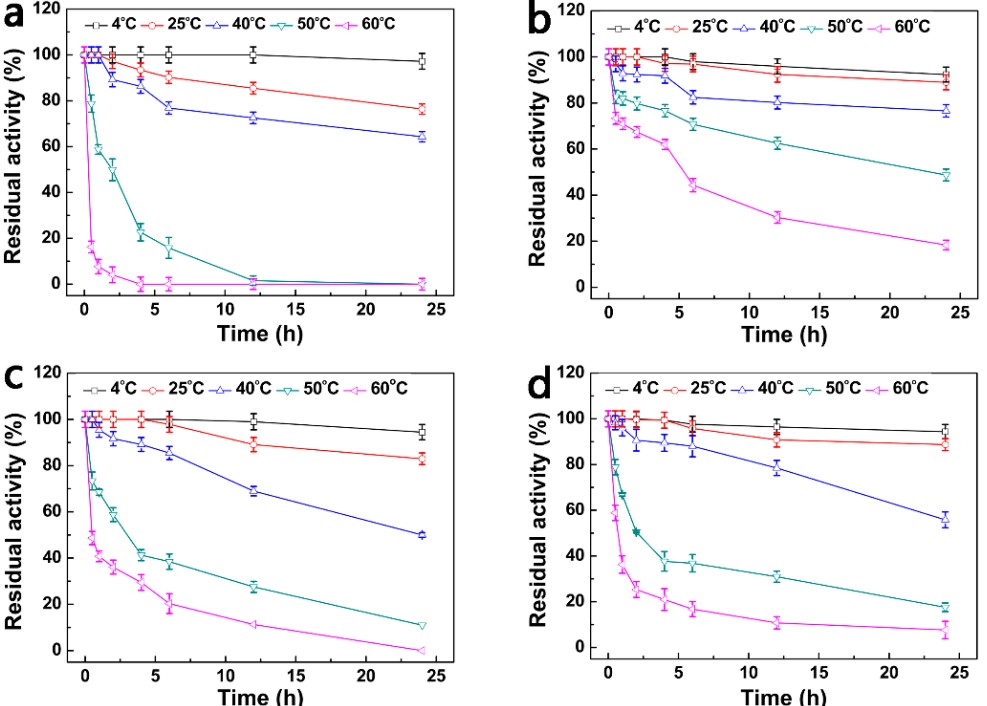

**Figure 5.** Thermostability of *Kp*-FDH (**a**), *Ti*-LDH (**b**), cofactor regeneration enzyme (*Kp*-FDH domains) (**c**) and carbonyl reductive amination (*Ti*-LDH domains) (**d**) in TLK.

### 2.3. Steady-State Kinetic Parameters

Using potassium cyclopropylglyoxylate and ammonium formate concentrations as variants, steady-state kinetic analysis was performed to evaluate the catalytic activities of the parent enzyme *Kp*-FDH and *Ti*-LDH, and *Kp*-FDH domains and *Ti*-LDH domains in

the bifunctional enzyme TLK. The kinetic parameters $K_m$, $k_{cat}$, and corresponding catalytic efficiencies ($k_{cat}/K_m$) are listed in Table 2 and shown in Figure S2. The results demonstrated that the fusion of *Ti*-LDH with *Kp*-FDH did not significantly affect either $K_m$ or $K_{cat}$ of *Ti*-LDH subunits and *Kp*-FDH in the TLK. However, in comparison with the parent enzymes, their constants were modified for two subunit activities of TLK. The $K_{cat}$ decrease of 9.61 times and the $K_m$ increase of 7.12 times only resulted in a 1.35 time-loss of $K_{cat}/K_m$ of *Ti*-LDH subunit activity in the TLK. The decrease in catalytic activities might be ascribed to the fact that some active sites of *Ti*-LDH subunits in the TLK were inaccessible for potassium cyclopropylglyoxylate. A similar phenomenon was also observed in the *Kp*-FDH subunits of TLK. Even so, the obtained kinetic parameters still allowed for its use in the biocatalytic asymmetric synthesis of (*S*)-cyclopropylglycine.

**Table 2.** Kinetic parameters of the parent enzymes and the bifunctional enzymes.

| | *Kp*-FDH | *Ti*-LDH | TLK | |
| --- | --- | --- | --- | --- |
| | | | *Kp*-FDH | *Ti*-LDH |
| $K_m$ (mM) | 10.15 | 2.35 | 34.00 | 0.33 |
| $k_{cat}$ ($S^{-1}$) | 75.31 | 919.77 | 56.10 | 95.74 |
| $k_{cat}/K_m$ ($S^{-1}$ $mM^{-1}$) | 7.42 | 391.39 | 1.65 | 290.11 |

*2.4. Preparation of Cyclopropylglyoxylic Acid*

Inexpensive cyclopropyl methyl ketone was oxidized with aqueous $KMnO_4$ at 50 °C in the presence of $K_2CO_3$ [10,13]. After the addition of all components, the mixture was mechanically stirred for an additional 3 h. Then, 3.5% $H_2O_2$ was added to remove the excess $KMnO_4$. After the removal of $MnO_2$ by the filtration, the filtrates were concentrated in vacuo to a solid residue. Recrystallization from $CH_3OH$ provided potassium cyclopropylglyoxylate as a nacreous solid. LC-MS analysis demonstrated that the fragment ion of potassium cyclopropylglyoxylate was $m/z$ 153.0, in line with the theoretical molecular weight (Figure S3). The $^1$H NMR (600 MHz, $D_2O$) spectrum (Figure S4) and $^{13}$C NMR (600 MHz, $D_2O$) spectrum (Figure S5) were well in line with published data [10].

*2.5. Proximity Effect Study*

To evaluate the influence of proximity effect on biocatalytic activity, a combination of *Ti*-LDH and *Kp*-FDH enzyme and bifunctional enzyme TLK-catalyzed bioconversion reaction was carried out concurrently according to the bioconversion process (Figure 2c). It was found that the bioconversion rate of potassium cyclopropylglyoxylate catalyzed by *E. coli* (TLK) was 2.1-fold higher than that by the combination of *E. coli* (*Ti*-LDH) and *E. coli* (*Kp*-FDH) (Figure 6), indicating that the biocatalytic activities of *Ti*-LDH module and *Kp*-FDH module in the bifunctional enzyme TLK in vivo were not impaired by the fusion. The result might be ascribed to the close proximity of the active sites in TLK. When using two separate strains (Figure 2a), the high mass transfer resistance led to a low biocatalytic efficiency, thus the bioconversion process was generally different from industrial application. However, when the bioconversion process was conducted in a single strain (Figure 2c), there was a lower transit time in which NADH produced by *Kp*-FDH subunit in TLK reached the *Ti*-LDH active sites in TLK, and the higher local concentration of the enzyme clearly reduced the mass transfer resistance and increased the probability of the enzyme undergoing subsequent biocatalytic reaction before escaping by diffusion [15,16]. This is especially important for preparative applications.

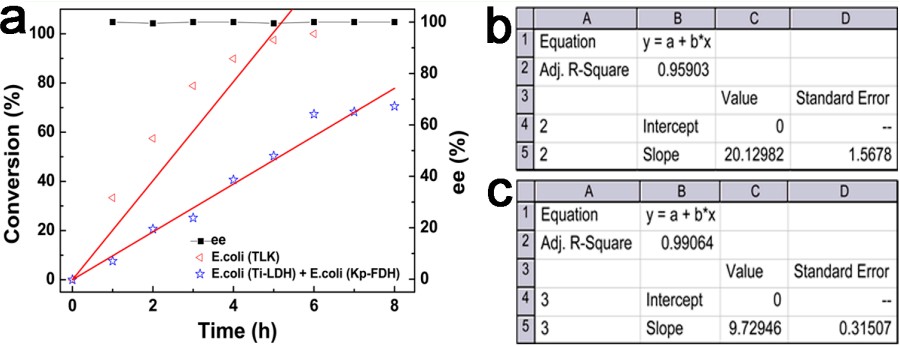

**Figure 6.** (**a**) Biocatalytic synthesis of (*S*)-cyclopropylglycine catalyzed by a combination of *E. coli* (*Ti*-LDH) and *E. coli* (*Kp*-FDH) and a self-sufficient whole-cell *E. coli* (TLK). The slopes of the straight lines represented the overall rate of (*S*)-cyclopropylglycine using a combination of *E. coli* (*Ti*-LDH) and *E. coli* (*Kp*-FDH) (**c**) and *E. coli* (TLK) (**b**), respectively.

### 2.6. NADH-Driven Biocatalytic Asymmetric Synthesis of (S)-Cyclopropylglycine

An efficient and robust biocatalyst should exhibit good biocatalytic performance. The NADH-driven biocatalytic system was first developed with the use of *E. coli* (TLK) for asymmetric amination synthesis of (*S*)-cyclopropylglycine and, concurrently, cofactor regeneration in situ. The biocatalytic reaction conditions including pH, temperature, cofactor concentration, and the mole ratio of ammonium formate to substrate were optimized. As shown in Figure 7, the optimized pH, temperature, cofactor concentration, and the mole ratio of ammonium formate to the substrate for TLK with cyclopropyl-glyoxylic acid potassium as substrate were found to be 8.0, 40 °C, 0.6 mM and 3, respectively. To fulfill the catalytic performance of the biocatalysts, all biocatalytic reactions were carried out in the optimum conditions in PBS solution.

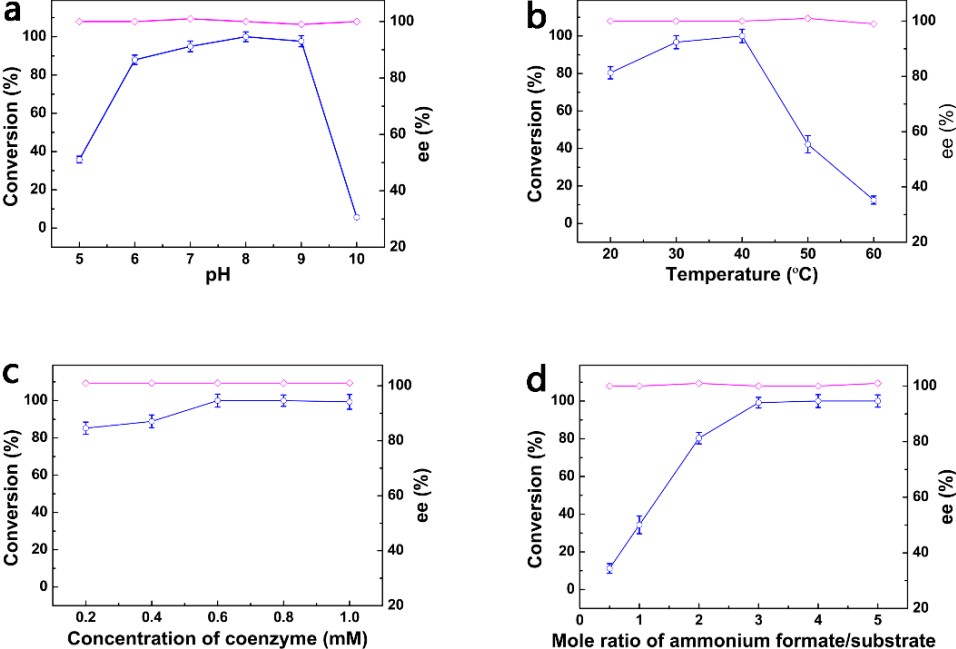

**Figure 7.** The effects of pH (**a**), temperature (**b**), the cofactor concentration (**c**) and the mole ratio of ammonium formate to substrate (**d**) on the biocatalytic performance of whole-cell biocatalysts.

To evaluate the practical applications of the developed biocatalytic system, we used potassium cyclopropylglyoxylate as a substrate for the asymmetric synthesis of (*S*)-cyclopropylglycine. The time course of the biocatalytic reaction suggested that a faster biotransformation rate was achieved with the increase in the amount of the biocatalysts or coenzyme used

(Figure 8). It was found that when the substrate loading was 100 g/L, the bioconversion reaction proceeded highly efficiently in the presence of very small amounts of cofactor, providing the desired (*S*)-cyclopropylglycine with 100% conversion after 3 h (Figure S6a) and an enantioselectivity of 100% ee, implying that TLK had good catalytic activity for cyclopropylglyoxylic acid potassium. Even when the *E. coli* (TLK) was reduced by half, the bioconversion reaction still completely proceeded to provide >99% ee and >99% conversion within 6 h. Under the same bioconversion conditions, the substrate loading was then increased to 120 g/L, and the bioconversion was completed easily within 6 h (Figure S6b). HPLC analysis was also employed to monitor the generated (*S*)-cyclopropylglycine during the bioconversion process (Figure S7). The retention time for (*S*)-cyclopropylglycine was about 2.5 min, while potassium cyclopropylglyoxylate did not display the corresponding absorption peak at 220 nm. The concentration of the generated (*S*)-cyclopropylglycine increased with the increase in the reaction time. After a reaction time of 6 h, the amount of (*S*)-cyclopropylglycine did not significantly increase, indicating that the bioconversion reaction finished. On the other hand, this also confirmed that ninhydrin could be used to monitor the bioconversion process. Notably, even if the amount of the coenzyme used was reduced by half in comparison with that of the previous batch, it still afforded >99% ee and >99% conversion within 8 h. Subsequently, an impressive substrate concentration (140 g/L) was used in the presence of 7.5 g/L biocatalysts and 0.6 mM cofactor. Strikingly, the bioconversion of the substrate still reached 100% within 12 h (Figure S6c). Even so, the bioconversion time was still far lower than the reported time (17 h) [13]. The space–time yield (sty) of (S)-cyclopropylglycine and the total turnover number (TTN), respectively, reached 377.3 g $L^{-1}$ $d^{-1}$ and 1365. For biocatalysis to be used in industrial production, whether it employs isolated enzymes or whole cells, it should meet the following targets: $\geq$100 g/L substrate loading, $\geq$98% conversion, $\geq$99% ee, and $\leq$24 h of reaction time [27]. Obviously, the developed biocatalytic process well met these requirements. Additionally, the biocatalysts could be lyophilized and stored conveniently over two months at $-80\,^{\circ}$C without obvious activity loss (Figure S8), thus being good for long-time storage. FT-IR analysis confirmed successful asymmetric reductive amination synthesis of (*S*)-cyclopropylglycine due to the disappearance of the carbonyl peak, along with the presence of the amino peak (Figure S9). To measure the ee of (*S*)-cyclopropylglycine, analysis on a CHIRALPAK® MA (+), 5 × 0.46 cm, 3 μm column (Daicel, Japan) was performed using 2 mM $CuSO_4$ as mobile phase at a flow rate of 0.5 mL/min at 25 $^{\circ}$C and detection by UV at 254 nm. The injection volume was 5 μL. The retention times for the (R)-cyclopropylglycine and (*S*)-cyclopropylglycine were 5.023 and 7.870 min, respectively, while the retention time for the synthesized (*S*)-cyclopropylglycine was 7.905 min without the presence of (R)-enantiomer, indicating that the ee of the synthesized (*S*)-cyclopropylglycine was >99% (Figure S10), in line with the reported result [13]. LC-MS, [1]HNMR and [13]C NMR spectrum further confirmed the structure of the synthesized (*S*)-cyclopropylglycine (Figures S11–S13). The optical rotation was +112.05 (c 10 mg/mL, 1 M HCl) at 589 nm and room temperature, in agreement with the optical rotation of the standard product (+111.9) under the same conditions.

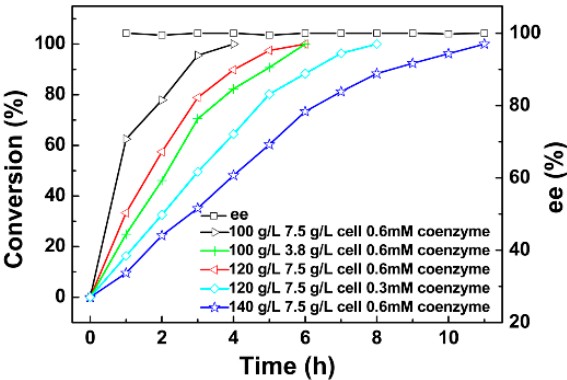

**Figure 8.** The time course of reductive amination of potassium cyclopropylglyoxylate to produce (*S*)-cyclopropylglycine in the presence of coenzyme at 40 °C and pH 8.0.

## *2.7. Continuous Synthesis of (S)-Cyclopropylglycine*

Recovery and reuse of biocatalysts are of significant importance. A dialysis method was developed to test the durability and efficiency of a continuous synthesis of (*S*)-cyclopropylglycine. In our experience, the bifunctional enzyme TLK is eminently recyclable. After each cycle, the dialysis tube containing *E. coli* (TLK) cell lysates is transferred into a new reaction medium, and directly employed again in the next cycle. As shown in Figure 9a, the bifunctional enzyme TLK maintained its activity up to 12 changes with >99.5% ee and over 95% conversion yield. During the 90-h operation, about 634.6 g of (*S*)-cyclopropylglycine was synthesized with 7.5 g lyophilized *E. coli* (TLK) being used. When (*S*)-cyclopropylglycine synthesis ceased, about 35% of both *Ti*-LDH and *Kp*-FDH activity in TLK still remained, as can be seen in Figure 9b. The significant decrease in the catalytic activity could be ascribed to the enzyme denaturation.

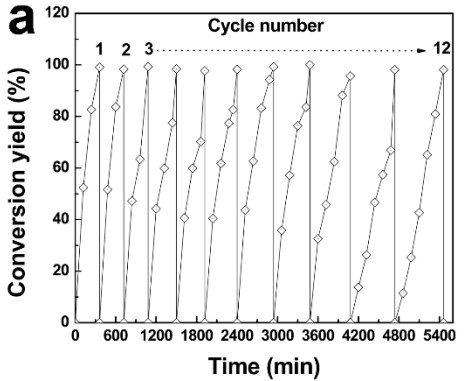
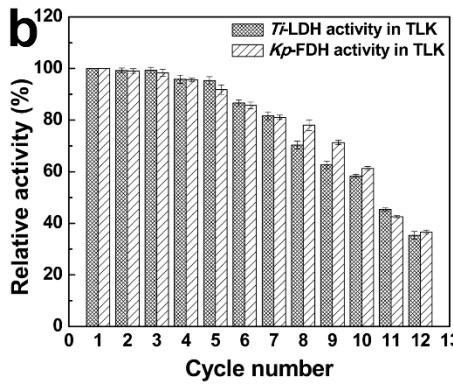

**Figure 9.** (**a**) Continuous synthesis of (*S*)-cyclopropylglycine. When the last cycle was completed, the dialysis tube containing *E. coli* (TLK) cell lysates was directly transferred to a new reaction medium, indicative of the start of the next one. The reaction was continued for 12 cycles. (**b**) Relative activity of *Ti*-LDH and *Kp*-FDH in TLK.

## 3. Experimental Section

### *3.1. Materials*

Competent cells *E. coli* BL21 (DE3) and DH5α were supplied by Tiangen (Beijing, China). Page Ruler Prestained Protein Ladder was purchased from ThermoFisher Scientific (Waltham, MA, USA). Isopropy-l-β-D-thiogalactopyranoside (IPTG), Kanamycin and all reagents for sodium dodecyl sulfate polyacrylamide gel electrophoresis (SDS-PAGE) were from Sigma (Shanghai, China). All other chemicals were of analytical grade and are commercially available. Ultrapure water was from the Millipore system (Bedford, MA, USA).

### 3.2. Construct and Expression Bifunctional Enzyme

Leucine dehydrogenase (*Ti*-LDH) from *Thermoactinomyces intermedius* (EC:1.4.1.9) and formate dehydrogenase (*Kp*-FDH) from *Komagataella pastoris* (EC:1.17.1.9) were chosen to construct the bifunctional enzyme, donated as TLK (linked at C-terminal of *Ti*-LDH with N-terminal of *Kp*-FDH) using GGGGS short peptide a flexible linker [25]. The amino acid sequence of TLK is listed in Figure S14. As a control, another bifunctional enzyme (KLT) was also constructed by linking the N-terminal of *Ti*-LDH with the C-terminal of *Kp*-FDH. Whole cloning procedures were performed by the manufacturer Taihe (Beijing, China). The genes encoding *Ti*-LDH, *Kp*-FDH, KLT, and TLK were synthesized in the pUC57 vector backbone, and then subcloned into pET28a, respectively. Corresponding expression vectors were transformed into *E. coli* BL21(DE3) to form the different recombinant strains, denoted as *E. coli* (*Ti*-LDH), *E. coli* (*Kp*-FDH), *E. coli* (KLT), and *E. coli* (TLK), respectively. Then, a 10 mL starter culture grown overnight from a single colony was utilized to inoculate 500 mL of Luria-Bertani (LB) broth (1% tryptone, 0.5% yeast extract, 1% NaCl) supplemented with 200 µg/mL of kanamycin in a 3 L conical flask. This culture was incubated at 37 °C and 150 rpm up to $OD_{600} \sim 0.8$. The temperature was then shifted to 25 °C, and enzyme expression was induced by the addition of 0.2 mM isopropyl-β-D-thiogalactoside (IPTG) for 48 h. The *E. coli* cells were pelleted by centrifugation ($10,000 \times g$, 5 min, 4 °C), washed with ice phosphate-buffered saline (PBS pH 7.4) buffer, and used immediately or placed at −80 °C for storage. The enzyme expression conditions were optimized under different induction temperatures (16–37 °C), IPTG concentrations (0.05–1.0 mM) and induction times (24–72 h). The expression level was evaluated by 12% SDS-PAGE analysis. Protein concentrations were estimated by the Bradford protein assay with bovine serum albumin as the standard [28].

### 3.3. Enzyme Activity Assay

The reductive amination activity in the parent enzyme *Ti*-LDH and bifunctional enzyme TLK were measured in PBS (pH 8.5) using a fixed concentration of enzyme, 10 pmol of potassium cyclopropylglyoxylate, and 0.1 pmol of NADH. Formation of NAD$^+$ was monitored at 340 nm ($\varepsilon_{NADH}^{340} = 6220 \ M^{-1} \ cm^{-1}$) using the cuvette port of a UV-visible spectrophotometer (Thermo Fisher Scientific, Waltham, MA, USA) [13]. The coenzyme regenerating activity in *Kp*-FDH and bifunctional enzyme TLK was carried out in PBS (pH 8.0) using a fixed concentration of enzyme, 100 mM sodium formate and 1 mM NAD$^+$. Reduction in NAD$^+$ was followed at 340 nm by spectrophotometry. A linear absorbance change for at least the initial 1 min was monitored, and that for the initial 5 s was used for the calculation. One unit of enzyme activity was defined as the amount of enzyme that catalyzed 1 µmol NADH or NAD$^+$ in 1 min.

The effect of pH (5.0 to 10.0) and temperature (20 to 60 °C) on enzyme activity was investigated by the standard enzyme activity analysis procedure. Furthermore, the thermostability of the enzyme was also evaluated by measuring enzyme residual activity after incubation at different temperatures (4, 25, 40, 50, 60 °C).

### 3.4. Steady-State Kinetic Assays

Kinetic parameters $K_m$ and $k_{cat}$ for potassium cyclopropylglyoxylate were determined using fixed concentrations of *Ti*-LDH or TLK, 1 mM NADH, excess ammonium formate and varying potassium cyclopropylglyoxylate concentration (0–50 mM). Additionally, Kinetic parameters $K_m$ and $k_{cat}$ for ammonium formate were measured using fixed concentrations of *Kp*-FDH or TLK, 1 mM NAD$^+$, and varying ammonium formate concentrations (0–50 mM). $K_m$ and $k_{cat}$ were calculated by fitting the enzyme activity via substrate concentrations using the Michaelis–Menten equation.

### 3.5. Preparation of Cyclopropylglyoxylate

Cyclopropylglyoxylate was synthesized according to the previously reported method [10,13]. Briefly, cyclopropylmethyl ketone (20 g, 237.8 mmol) and sodium carbonate (290 mg,

2.72 mmol) were dissolved in water (132 mL) and heated to 50 °C, followed by dropwise addition of 1.2 L KMnO$_4$ aqueous solution (39.6 g, 250.4 mmol) over 10 h. The reaction was maintained for another 4 h at 50 °C. After the removal of excess KMnO$_4$ using H$_2$O$_2$, the mixture was filtered to remove MnO$_2$ black precipitate. The filtrate was then concentrated into a solid residue. Recrystallization from methanol provided the desired white product (potassium cyclopropylglyoxylate).

### 3.6. Proximity Effect Study

To evaluate the proximity effect, two sets of whole-cell biocatalysts, (i): a combination of *E. coli* (*Ti*-LDH) and *E. coli* (*Kp*-FDH), (ii): *E. coli* (TLK), were used to catalyze the bioconversion, and the reaction rates were determined and compared with each other. Each set of biocatalysts had approximately equal U for reductive amination and coenzyme regeneration activities. A reaction mixture (10 mL) containing potassium cyclopropylglyoxylate (0.61 g, 4 mmol), ammonium formate (0. 76 g, 12 mmol), NAD$^+$ (4 mg) and the lyophilized cells (75 mg), was incubated at 40 °C. At different times, aliquots of reaction solution were withdrawn to quantify the formed (*S*)-cyclopropylglycine by ninhydrin colorimetric assay using Uv-vis spectrophotometer [29,30].

### 3.7. Reductive Amination of Cyclopropylglyoxylic Acid with the Self-Sufficient Whole-Cell Biocatalysts

To fully exhibit the biocatalytic performance of the whole-cell biocatalysts, the biocatalytic conditions including temperature (20–60 °C), pH (5.0–10.0), the mole ratio of ammonium formate to substrate (1–5) and the concentration of NAD$^+$ (0.2–1.0 mM) were optimized by adding 7.5 g/L of lyophilized cells. Then, 2,4-dinitrophenylhydrazine and Ninhydrin were used to preliminarily detect the decrease in substrate concentration and the increase in product concentration [29,30], respectively. The effect of long-period storage on biocatalytic activity was also investigated. The following procedure was recommended for the preparation of (*S*)-cyclopropylglycine. Typical bioconversion reaction catalyzed by the self-sufficient whole-cell biocatalysts was performed in 500 mL of PBS (100 mM, pH 8.0) reaction medium containing 0.8 M potassium cyclopropylglyoxylate, ammonium formate (2.4 M), 0.6 mM NAD$^+$ and 7.5 g/L of lyophilized cells in a 2 L conical flask with continuous shaking at 125 rpm at 40 °C. pH 8.0–8.5 was maintained by adding 5 M NaOH solution. At the end of the reaction, trichloroacetic acid was added to the reaction mixture to precipitate the proteins. The mixture was heated to 80–100 °C, and then cooled to room temperature. After that, the activated carbon filter was carried out to remove the formed precipitates. The filtrate was applied to a cation exchange resin 001 × 7 and eluted with 14% ammonium hydroxide. The eluent solution was concentrated by a rotary evaporator, giving (*S*)-cyclopropylglycine.

To monitor the biotransformation process, 20 mL of the reaction mixture was withdrawn at regular intervals and treated according to the above-mentioned method. The collected eluent solution from cation exchange resin was concentrated, diluted with borate buffer (pH 9.5, 0.05 mol/L), and further filtrated through a 0.2 µm membrane. Then, 20 µL of the filtrate was applied to Agilent Technologies G1314F HPLC system with a C18 analytical column (4.6 mm × 250 mm, 5 µm, Dalian Elite Analytical Instruments Co., Ltd., Dalian, China), while the column temperature was maintained at 30 °C, and the mobile phase was 5% acetonitrile/95% water with a flow rate of 1.0 mL/min at 220 nm according to the modified method [13].

The enantiomeric excess (EE) of (*S*)-cyclopropylglycine was determined by HPLC with a CHIRALPAK® MA (+) (MAP0CC-WD004) chiral column (0.46 cm I.D. × 5 cm L × 3 µm). The column temperature was maintained at 25 °C with a flow rate of 0.5 mL/min at 254 nm. The mobile phase was 2 mM CuSO$_4$, and the injection volume was 5 µL.

$^1$H and $^{13}$CNMR spectra were recorded at 600 MHz Bruker Ascend III HD with D$_2$O as solvent. LC-MS was carried out on Agilent 6470 Triple Quad LC MS (Agilent Technologies Inc., Santa Clara, CA, USA). Fourier transform infrared (FT-IR) spectra of the compounds

were measured on a Nicolet 6700 iS5 (Thermo Fisher Scientific, Waltham, MA, USA) in the wavenumber ranging from 4000 to 500 $cm^{-1}$ with a resolution of 4 $cm^{-1}$.

### 3.8. Continuous Synthesis of (S)-Cyclopropylglycine

According to the previously reported method [31], a continuous biocatalytic process for synthesizing (*S*)-cyclopropylglycine with the use of *E. coli* (TLK) as a biocatalyst was developed. Firstly, the *E. coli* (TLK) cell lysates were transferred into a dialysis tube and soaked in PBS (100 mM, pH 8.0) reaction medium containing 0.65 M potassium cyclopropylglyoxylate, ammonium formate (2.0 M), and 0.6 mM NADH in a total volume of 1 L. The dialysis tube containing *E. coli* (TLK) cell lysates was incubated at 40 °C and periodically transferred to a fresh reaction medium containing the same composition.

### 3.9. Statistical Analysis

Each experiment was conducted in triplicate. All values obtained from the experiments were presented as the mean ± standard deviation (SD). Statistical analysis of data was carried out by SPSS (version 16.0) with one-way analysis of variance (ANOVA). A value of $p < 0.05$ was considered statistically significant.

### 4. Conclusions

In conclusion, we developed a practical NAD(H)-driven biocatalytic system suitable for the asymmetric synthesis of (*S*)-cyclopropylglycine. By integrating *Ti*-LDH and well-known NADH-dependent *Kp*-FDH, the desired biocatalytic system was constructed, which made it simple to consolidate two reactions into the eco-efficient one-pot biocatalytic process by using the constructed bifunctional enzyme. This biocatalytic system proceeded with impressive substrate loading to synthesize (*S*)-cyclopropylglycine with high STY and >99% ee, exhibiting excellent durability and biocatalytic efficiency in a continuous synthesis of (*S*)-cyclopropylglycine. Moreover, the devised strategy can be used as an all-purpose biocatalyst for asymmetric reduction reactions involving in situ coenzymes such as reductive amination, C=O and C=C reduction. Thus, the biocatalytic system opens a new horizon for the biosynthesis of (*S*)-cyclopropylglycine.

**Supplementary Materials:** The following supporting information can be downloaded at: https://www.mdpi.com/article/10.3390/catal14050321/s1, Figure S1: Asymmetric synthesis of (*S*)-cycloprop-ylglycine catalyzed by whole-cell *E. coli* (KLT) (a), a combination of the two native enzymes *E. coli* (*Ti*-LDH) and *E. coli* (*Kp*-FDH) (b), and *E. coli* (TLK) (c); Figure S2: Steady-state kinetic of activities for different recombinant enzymes. Figure S3: The LC-MS of potassium cyclopropylglyoxylate; Figure S4: $^1$H NMR of potassium cyclopropylglyoxylate; Figure S5: $^{13}$C NMR of potassium cyclo-propylglyoxylate; Figure S6: Asymmetric synthesis of (*S*)-cyclopropylglycine catalyzed by whole-cell *E. coli* (TLK) in the presence of 7.5 g/L biocatalysts and 0.6 mM cofactor at different concentration of potassium cyclopropylglyoxylate; Figure S7: HPLC analysis of the generated (*S*)-cyclopropylglycine catalyzed by *E. coli* (KLT) at 120 g/L potassium cyclopropylglyoxylate in the presence of 7.5 g/L biocatalysts and 0.6 mM cofactor; Figure S8: The influence of storage time on the biocatalytic activity of the bifunctional enzyme TLK. The biocatalytic process was conducted under standard conditions; Figure S9: FT-IR spectra of cyclopropyl methyl ketone, potassium cyclopropylglyoxylate and (*S*)-cyclopropylglycine; Figure S10: Determination of optical purity of the products by HPLC; Figure S11: The LC-MS of (*S*)-Cyclopropylglycine; Figure S12: $^1$HNMR of (*S*)-Cyclopropylglycine; Figure S13: $^{13}$C NMR of (*S*)-Cyclopropylglycine; Figure S14: The amino acid sequence of TLK.

**Author Contributions:** Q.T., S.L. and W.L. designed the experiments; Q.T., S.L. and L.Z. performed the experiments; L.S. and J.X. analyzed the data; Q.T., S.L. and W.L. wrote the paper. All authors have read and agreed to the published version of the manuscript.

**Funding:** This research was funded by Municipal Science and Technology Committee of Chongqing: cstc2021jcyj-msxmX0212.

**Data Availability Statement:** Data are contained within the article and Supplementary Materials.

**Acknowledgments:** This work was supported by School of Pharmacy, Chongqing Medical University.

**Conflicts of Interest:** The authors declare no conflict of interest.

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
