# Peer review of "Facile Asymmetric Syntheses of Non-Natural Amino Acid (S)-Cyclopropylglycine by the Developed NADH-Driven Biocatalytic System"

_catalysts, doi:10.3390/catal14050321_

Round 1
Reviewer 1 Report
Comments and Suggestions for Authors
A bifunctional enzyme preparation has been used for the stereoselective synthesis of (S)-2-cyclopropyl-glycine. The approach consists of a reductive amination where enzymatic cofactor recycling is required to shift the equilibrium towards amino acid synthesis.
The introduction provides a comprehensive and interesting state of the art to the produce the target chiral amino acid where biocatalysis is identified as an ideal technology to achieve the aim of the project, I have really enjoyed it. Then, the research aims are clearly presented to later disclosed the main study achievements such as enzyme construction, biotransformation optimization, kinetic analyses, and recycling studies. Overall, this is a really nice piece of work that deserves publication in Catalysts with minor modifications. Some minor items:
- Page 2, line 47: “racemic methyl” instead of “racmethyl”
- Page 2, lines 51-52: “Meanwhile, enzyme-catalyzed kinetic resolution was carried out to improve the product 51 stereoselectivity [10,12], but giving < 50% yield in most cases” References 10 and 12 deals with the design of classical kinetic resolution using hydrolases (papain and acylases, respectively) that has the inherent limitation of <50% yield in enantiopure products, so the sentence must be re-written considering this fact.
- Page 7, line 199: It is unclear how MnO2 was removed for amino acid synthesis, I assume that a filtration is done (indicate washing solvent), and for the ta reason the filtrates are collected for subsequent product recrystallization. Yes indeed, see page 13 (line 363).
- Page 8, line 226: “asymmetric” instead of “symmetric”
- Page 9, lines 250, 261 & 281: due to the inherent limitation of the analytical technique, I would say “>99% ee” instead of “100% ee” (same for line 434 of page 13; 99.5% ee).
- Page 9, line 269: Remove bold font for reference [28].
- Page 11, line 346: As far as I have seen temperature screening is between 10 and 60 °C (not 20-80 °C)
- Experimental section. It would be interesting to explain in which vial type and volume the biotransformations were performed since only the shaking speed is claimed.
- ESI, legends Figure S1, S6 & S7: Indicate the PBS composition and pH
- ESI, legend Figure S4, S5, S12 & S13: Indicate the deuterated solvent and spectrometer frequenccy together with NMR signal description and identification.
-
Author Response
Manuscript title: Facile Asymmetric Syntheses of Non-natural Amino Acid (S)-cyclopropylglycine by Developed NADH-Driven Biocatalytic System
Manuscript ID: catalysts-2987761
Referee 1:
Comments to the Author:
************
A bifunctional enzyme preparation has been used for the stereoselective synthesis of (S)-2-cyclopropyl-glycine. The approach consists of a reductive amination where enzymatic cofactor recycling is required to shift the equilibrium towards amino acid synthesis.
The introduction provides a comprehensive and interesting state of the art to the produce the target chiral amino acid where biocatalysis is identified as an ideal technology to achieve the aim of the project, I have really enjoyed it. Then, the research aims are clearly presented to later disclosed the main study achievements such as enzyme construction, biotransformation optimization, kinetic analyses, and recycling studies. Overall, this is a really nice piece of work that deserves publication in Catalysts with minor modifications. Some minor items:
√Responses: Thank you very much for recognizing our work.
- Page 2, line 47: “racemic methyl” instead of “racmethyl”
√Responses: Thank you very much for your suggestions. We have corrected the word. Please see the main text.
- Page 2, lines 51-52: “Meanwhile, enzyme-catalyzed kinetic resolution was carried out to improve the product 51 stereoselectivity [10,12], but giving < 50% yield in most cases” References 10 and 12 deals with the design of classical kinetic resolution using hydrolases (papain and acylases, respectively) that has the inherent limitation of <50% yield in enantiopure products, so the sentence must be re-written considering this fact.
√Responses: Thank you very much for your suggestions. We have re-written this sentence. Please see the main text.
- Page 7, line 199: It is unclear how MnO2 was removed for amino acid synthesis, I assume that a filtration is done (indicate washing solvent), and for the ta reason the filtrates are collected for subsequent product recrystallization. Yes indeed, see page 13 (line 363).
√Responses: Thank you very much for your suggestions. Yes, you are right. We have revised this sentence. Please see the main text.
- Page 8, line 226: “asymmetric” instead of “symmetric”
- Page 9, lines 250, 261 & 281: due to the inherent limitation of the analytical technique, I would say “>99% ee” instead of “100% ee” (same for line 434 of page 13; 99.5% ee).
- Page 9, line 269: Remove bold font for reference [28].
√Responses: Thank you very much for your suggestions. We have corrected these inappropriate expressions. Please see the main text.
- Page 11, line 346: As far as I have seen temperature screening is between 10 and 60 °C (not 20-80 °C)
√Responses: Thank you very much for your suggestions. Indeed, the temperature screening is between 20 and 60 °C as evidenced by Line 152 and Fig 4a-d. Please see the main text.
- Experimental section. It would be interesting to explain in which vial type and volume the biotransformations were performed since only the shaking speed is claimed.
√Responses: Thank you very much for your suggestions. We have added the vial type and the biotransformation volume. Please see the main text.
- ESI, legends Figure S1, S6 & S7: Indicate the PBS composition and pH
√Responses: Thank you very much for your suggestions. We have indicated PBS composition and pH value in legends Figure S1, S6 & S7. Please see the supporting information.
- ESI, legend Figure S4, S5, S12 & S13: Indicate the deuterated solvent and spectrometer frequency together with NMR signal description and identification.
√Responses: Thank you very much for your suggestions. We have indicated the deuterated solvent and spectrometer frequency, and NMR signal description and identification in legend Figure S4, S5, S12 & S13. Please see the supporting information.
Reviewer 2 Report
Comments and Suggestions for Authors
I had to review a very interesting manuscript. The synthesis of chiral non-racemic amino acids is of great importance in organic chemistry, with particular emphasis on compounds with desired biological activity. The synthesis of non-racemic compounds remains a great challenge for chemists. Therefore, methods for the effective synthesis of non-racemic unnatural amino acids, which are valuable building blocks, are being sought. Unfortunately, the available classical methods have numerous disadvantages, including metal contamination. Biocatalytic methods are an attractive alternative to the classical approach. Unfortunately, in the case of processes catalyzed by isolated enzymes, it is necessary to regenerate co-factors, compounds characterized by high prices. Enzyme systems are used for this purpose. However, due to diffusion problems and incompatibility, individual enzymes achieve their highest activity under different conditions. Currently, the latest approach to avoid these limitations is to design and obtain fusion biocatalysts combining the catalytic activity of individual enzymes. The authors successfully managed to obtain a fusion enzyme that catalyzes in its active domains both the reductive amination reaction and the regeneration of the co-factor necessary for the enzyme's operation. The approach used allowed to obtain a construct with high operational stability and catalytic activity in the synthesis of S-cyclopropylglycine. The obtained amino acid was comprehensively analyzed. The experiments designed by the authors were carefully thought out and executed. The results obtained were correctly interpreted and presented in a clear way. I have a few minor comments which should make the work accepted
I would like to ask you to describe the NMR spectra, not just show scans of the spectra, and compare them with literature data for known compounds. Giving the optical rotation for an amino acid. I would also ask you to include the amino acid sequence of the enzymes, which will enable interested readers to repeat the presented results and create their own fusion enzymes without having to search for sequences in dedicated databases.
Author Response
Manuscript title: Facile Asymmetric Syntheses of Non-natural Amino Acid (S)-cyclopropylglycine by Developed NADH-Driven Biocatalytic System
Manuscript ID: catalysts-2987761
Referee 2:
Comments to the Author:
************
I had to review a very interesting manuscript. The synthesis of chiral non-racemic amino acids is of great importance in organic chemistry, with particular emphasis on compounds with desired biological activity. The synthesis of non-racemic compounds remains a great challenge for chemists. Therefore, methods for the effective synthesis of non-racemic unnatural amino acids, which are valuable building blocks, are being sought. Unfortunately, the available classical methods have numerous disadvantages, including metal contamination. Biocatalytic methods are an attractive alternative to the classical approach. Unfortunately, in the case of processes catalysed by isolated enzymes, it is necessary to regenerate co-factors, compounds characterized by high prices. Enzyme systems are used for this purpose. However, due to diffusion problems and incompatibility, individual enzymes achieve their highest activity under different conditions. Currently, the latest approach to avoid these limitations is to design and obtain fusion biocatalysts combining the catalytic activity of individual enzymes. The authors successfully managed to obtain a fusion enzyme that catalyses in its active domains both the reductive amination reaction and the regeneration of the co-factor necessary for the enzyme's operation. The approach used allowed to obtain a construct with high operational stability and catalytic activity in the synthesis of S-cyclopropylglycine. The obtained amino acid was comprehensively analysed. The experiments designed by the authors were carefully thought out and executed. The results obtained were correctly interpreted and presented in a clear way. I have a few minor comments which should make the work accepted.
√Responses: Thank you very much for recognizing our work.
I would like to ask you to describe the NMR spectra, not just show scans of the spectra, and compare them with literature data for known compounds. Giving the optical rotation for an amino acid. I would also ask you to include the amino acid sequence of the enzymes, which will enable interested readers to repeat the presented results and create their own fusion enzymes without having to search for sequences in dedicated databases.
√Responses: Thank you very much for your suggestions. We have described the NMR signal description and identification, and provided the optical rotation and the amino acid sequence of the TLK. Please see the main text and the supporting information.
